# Cow and Human Milk-Derived Exosomes Ameliorate Colitis in DSS Murine Model

**DOI:** 10.3390/nu12092589

**Published:** 2020-08-26

**Authors:** Shimon Reif, Yaffa Elbaum-Shiff, Nickolay Koroukhov, Itamar Shilo, Mirit Musseri, Regina Golan-Gerstl

**Affiliations:** 1Department of Pediatrics, Hadassah-Hebrew University Medical Center, Jerusalem 9112001, Israel; shimon@hadassah.org.il (S.R.); yaffa.elbaumshif@mail.huji.ac.il (Y.E.-S.); kor_nickolay@hotmail.com (N.K.); shiloi@hadassah.org.il (I.S.); miritm@hadassah.org.il (M.M.); 2Nutrition and Dietetic Department, Tel Aviv Medical Center, Tel Aviv 6423906, Israel

**Keywords:** exosomes, miRNA, milk, colitis

## Abstract

The aim of this study was to investigate the therapeutic effect of cow and human milk derived exosomes (MDEs) on colitis. We used gavage administration of fluorescent labeled MDEs to track their localization patterns in vivo and studied their therapeutic effect on colitis in a dextran sulfate sodium (DSS)-induced colitis model. MDEs attenuated the severity of colitis induced by DSS and statistically reduced the histopathological scoring grade and shortening of the colon. Likewise, treatment with MDEs reduced the expression of interleukin 6 and tumor necrosis factor-α. Moreover, miRNAs highly expressed in milk, such as miRNA-320, 375, and Let-7, were found to be more abundant in the colon of MDE-treated mice compared with untreated mice; contrastingly, the expression of their target genes, mainly DNA methyltransferase 1 (DNMT1) and DNMT3 were downregulated. Furthermore, the level of TGF-β was upregulated in the colon of MDE-treated mice. We demonstrated that MDEs have a therapeutic and anti-inflammatory effect on colitis, involving several complementary pathways in its mechanism of action. The therapeutic effects of MDEs might have implications for the possible addition of MDEs as a nutrient in enteral nutrition formulas for patients with inflammatory bowel disease.

## 1. Introduction

Inflammatory bowel disease (IBD) is a chronic relapsing inflammatory disease of the gastrointestinal tract. IBD is characterized by two major phenotypes: ulcerative colitis and Crohn’s disease. Both of them usually characterize by severe occasionally, abdominal pain, bloody diarrhea and weight loss. Although they share many similarities, there are key differences between ulcerative colitis and Crohn’s disease. Ulcerative colitis is limited to the colon whereas Crohn’s disease can occur anywhere between the mouth and the anus. In Crohn’s disease, there are healthy parts of the intestine mixed in between inflamed areas, which named skipped areas. Ulcerative colitis, on the other hand, is a continuous inflammation of the colon without involving the small intestine. Ulcerative colitis is a superficial inflammation which affects the mucosal layer of the colon while in Crohn’s disease the inflammation is transmural and can affects all intestine layers [1]. Multiple factors are involved in the pathogenesis of IBD. It has been postulated that the etiology of IBD involves a complex interaction between the immune response, gastrointestinal microbiota, genetic susceptibility and environmental factors. Whereas, the prevalence of IBD has been increasing, no cure has been so far identified for the disease. Current and emerging therapies for the treatment of IBD have been mainly focused on immune-modulation. Meanwhile, researchers are trying to find new treatments to eliminate the disease and improve its complications. Patients with IBD frequently experience relapses, and even with current more effective medical therapies, such as biologic agents long term remission cannot be achieved; thus, this is the reason for the need to explore newer and more effective treatments. Currently, the role of exclusive enteral feeding as the first line therapy in IBD has been well established, mainly in pediatric practice [2].

Exosomes are nanoparticles containing a wide variety of proteins, miRNAs, lipids and mRNAs. Exosomes are known to transfer their cargo to recipient cells, thus serving as extracellular messengers to mediate cell–cell communication [3]. Milk derived exosomes (MDE) protect miRNAs and proteins from degradation in the digestive tract and thus can transfer them into the intestine facilitating their uptake and endocytosis.

As with other groups, we found that mammalian milk contains high concentrations of exosomes carrying beneficial, immune-related miRNAs, such as miRNA-148, representatives of the Let7 family, and miRNA-200 [4,5]. We previously demonstrated that following incubation of MDEs with intestinal epithelial cells, miRNAs were up taken by cells [6]. Furthermore, MDEs have been reported to modify the expression of target genes, such as DNA methyltransferase 1 (DNMT1) [4], promoting proliferation and differentiation of colon epithelial cells [6,7].

Accordingly, they have recently emerged as a potential new treatment modality for autoimmune and inflammatory diseases, such as rheumatoid arthritis [8]. Exosomes have been found in physiological fluids, such as bronchoalveolar lavage [9], serum [10], urine [11], and milk [12]. Mammalian milk, however, is known to contain the highest amount of exosomes and is thus considered a rich source for their isolation [13]. Several studies have provided evidence for the promising protective effects of exosomes derived from mesenchymal stem and dendritic cells in colitis [14,15]. Previous studies, including our own, have found that mammalian milk contains a high concentration of exosomes carrying beneficial miRNAs, such as miRNA-148 [4] and immune-suppressive cytokines, such as transforming growth factor beta (TGF-β) [16]. Indeed, previous studies have shown that oral delivery of vesicles isolated from cow milk ameliorates the symptoms of experimental arthritis [8].

Notably, IBD is a complex disorder resulting from a dysregulated immune response in the gut. Current and emerging therapies for the treatment of IBD have mainly focused on immunosuppression. These therapies are targeted to three main processes of colitis; immunological (immune cell trafficking and activation), restoration of normal intestinal microbial flora, and recovery of the mucosal damage. MDEs have the potential to regulate the above-mentioned processes involved in the pathogenesis of colitis, as they are enriched in immune-related miRNAs [5] or TGF- β [16], which protects the intestine from damage, contributing to the restoration of the mucus [17]. In addition milk derived extracellular vesicles alter the microbiome composition [18].

Based on the composition and biological properties of MDEs, we investigated here whether MDEs could be used for the treatment of dextran sulfate sodium (DSS)-induced colitis in a murine model.

## 2. Materials and Methods

### 2.1. Exosomes Isolation

From human milk: Exosomes were isolated by sequential ultracentrifugation and filtration. The milk samples were fractionated by centrifugation at 6500× *g* for 30 min at 4 °C. Two fractions were obtained from each sample: fat and skim milk. The exosomes were isolated from the skim fraction. The skim milk was centrifuged at 12,000× *g* for 1 hour at 4 °C to remove debris. The skim was then passed through 0.45 μm and 0.22 μm filters to remove residual debris. The filtered supernatant was centrifugated at 135,000× *g* for 90 min at 4 °C to pellet the exosomes.

From cow milk: Given that cow’s milk contains considerably more casein than human milk, we added small changes to the protocol described before. Following filtrations, the skim was centrifugated at 70,000× *g* for 30 min at 4 °C to discard casein. The supernatant was centrifugated at 135,000× *g* for 90 min at 4 °C to pellet the exosomes.

The exosomes pellet was left over night in Phosphate-buffered saline (PBS) for 4 °C to dissolve the exosomes. MDEs were filtered (0.22 μm). The protein content of the exosome’s preparation was measured by a Bicinchoninic Acid (BCA) protein assay.

### 2.2. Electron Microscopy

MDEs were analyzed by electron microscopy as previously described [6] and were examined using a Jem-1400 Plus transmission electron microscope (Jeol, Peabody, MA, USA).

### 2.3. Nanoparticle Analysis

Nanoparticle tracking analysis was performed using a NS300 nanoparticle analyzer (NanoSight, Malvern, Worchestershire, UK), which was used to measure the size distribution of MDEs. Briefly, PBS-suspended MDEs were loaded into the sample chamber of the Nanosight unit, a blue laser source at 488 nm was applied to the diluted MDE suspension, and a video was recorded for 60 s at a frame rate of 24.98 fps. The movement of particles was analyzed using NTA software.

### 2.4. Dynamic Light Scattering

We performed dynamic light scattering (DLS) and zeta potential determinations using a Zetasizer nanoseries instrument (*λ* = 532 nm laser wavelength; Malvern Nano-Zetasizer, Worchestershire, UK). The MDE size data was referred to the distribution of scattering intensity (z average).

### 2.5. Extraction of Total RNA

#### 2.5.1. From Exosomes

Trizol reagent (INVITROGEN, Carlsbad, CA, USA) was added to the MDEs pellet [6]. Extraction of RNA was performed as previously described [6].

#### 2.5.2. From Colon Tissue

Stainless steel beads (5 mm mean diameter) and 300 µL of Trizol reagent was added to a section of the colon tissue. The tissue was homogenized on the TissueLyser5. The sample was centrifuge briefly to ensure that all the tissue debris is on the bottom of the tube. The RNA isolation was continued from the supernatant using Zymo Direct-zol RNA MiniPrep Kit (Zymo Research, Irvine, CA, USA) according to the manufacture protocols.

RNA quantity and quality were assessed by measuring the absorbance at different wavelengths using a NanoDrop spectrophotometer (Waltham, MA, USA) of the RNA samples.

### 2.6. MiRNA Detection by qReal Time-PCR (qRT-PCR)

500 ng of total RNA extracted from colon tissue and 100 ng of RNA extracted from MDEs were used to prepare cDNA. The qScript micro-RNA cDNA Synthesis Kit (Quantabio, Beverly, MA, USA) was used to reverse-transcribe the RNA to cDNA, according to the manufacturer’s instructions, and the resulting cDNA was used to assess the expression of miR-148a, miR-320, miR-375, Let-7a and RNU6. RNU6 was used as reference for normalization of miRNA expression levels. The PerfeCTa SYBR Green SuperMix (Quantabio, Beverly, MA, USA) was used together with Quantabio micro-RNA qPCR primers for the miR-148a-3p (HSMIR-0148A-3P), miR-320a (HSMIR-0320A), miR-375 (HSLET-0007A-5P), Let-7a (HSLET-0007A-5P) and RNU6 (HS-RNU6) obtained from Quantabio (Beverly, MA, USA.). The qRT-PCR was run using a two-step cycling protocol as previously described [4]. Normalization and relative expression level calculation was carried out using the 2^ (-ΔΔ CT) method.

### 2.7. mRNA Detection by qReal Time-PCR

For the quantification of mRNA complementary, cDNA was generated using the high capacity RNA-cDNA kit (Applied Biosystems, Foster City, CA, USA) according to the manufacturer’s instructions. Total RNA isolated from colon tissue and cells (1 µg) was used to generate cDNA. The cDNA was subjected to qRT-PCR. The mRNA levels were measured using qRT-PCR with master mix (Fast qPCR SyGreen Blue Mix, PCR Biosynthesis, Wayne, PA, USA).

The PCR reaction steps were performed as previously described [6]. For primers list see online Appendix A.

### 2.8. Immunoblotting

Protein lysates from colon tissue and MDE were separated and transferred onto a polyvinylidene difluoride (PDVF) membrane. The membranes were probed with antibodies, and detected using enhanced chemiluminescence detection. Primary antibodies were as follows: anti CD81 (1:1000; Cosmo Bio, Tokyo, Japan), anti HSP70 (1:1000; SBI System Biosciences, Palo Alto, CA, USA), anti-CD9 (1:1000; SBI System Biosciences, Palo Alto, CA, USA), rabbit anti TGF-β1 (Abcam, Cambridge, MA, USA), and rabbit anti β-actin (Abcam, Cambridge, MA, USA). The secondary antibody was horseradish per-oxidase (HRP)-conjugated goat anti-mouse or anti-rabbit (1:3000; Cell Signaling Technology, Danvers, MA, USA). Quantification was performed using NIH-Image software (Rasband, W.S., ImageJ, U. S. National Institutes of Health, Bethesda, MD, USA, https://imagej.nih.gov/ij/, 1997–2018).

### 2.9. Dir-Labeled MDEs

MDEs were incubated with 1 µM fluorescent lipophilic tracer 1,1-dioctadecyl-3,3,3,3-tetramethylindotricarbocyanine iodide (DiR) (INVITROGEN, Carlsbad, CA, USA) at 37 °C for 15 min. Following incubation, PBS was added to the mix, which was then centrifuged at 100,000× *g* for 60 min. Labeled exosomes were pelleted, whereas unbound label was discarded. The MDE uptake experiments were performed twice.

### 2.10. Dextran Sulfate Sodium-Induced Colitis Model in Mice

Colitis was induced in 8-weeks male Balb/c mice (Envigo RMS, Israel) using 5% DSS (dextran sulfate sodium salt, colitis grade (36,000–50,000) (MP Biomedicals, LLC, France) provided for 7 days in the drinking water. To assess the effects of DSS, potential behavioral alterations and changes in body weight were checked daily. At 1 week. from starting DSS, the water was changed to regular water. Mice received 50 mg/kg human or cow MDEs in 200 μL PBS orally by gavage for 6 days. At the end of the experiment, mice were sacrificed and their colon was removed, analyzed for its outward appearance and measured for length. After removing cecum and adipose tissue, the colon was cut into 3 parts: proximal, middle, and distal (each part = 2 cm). The distal part was fixed in 4% formalin solution, followed by paraffin embedding, staining with hematoxylin and eosin (H&E), and examined under a light microscope. For histological scoring the following parameters were evaluated: inflammation (0–2), lymphocyte infiltration (0–2), erosion (0–2) and crypt loss (0–3). The total score for a given section is the sum of those parameters and the maximum score is 9. In addition, cell differentiation, hemorrhages, fibrin deposition, and lesion distribution were also evaluated. The proximal and middle part of the colon were frozen in liquid nitrogen and stored until use at −80 °C for protein, gene and miRNA expression analysis. For protein analysis, stainless steel beads (5 mm mean diameter) and 200 µL of RIPA buffer with proteinase inhibitor cocktail was added to a section of the colon tissue. The tissue was homogenized on the TissueLyser. The sample was centrifuge briefly to ensure that all the tissue debris is on the bottom of the tube. The protein content was measured by a BCA protein assay (Thermo). Gene and miRNA expression analysis in colon tissue were performed following RNA extraction. MDE exosomes were administrated in three separate DSS induced colitis experiments (two with human MDE and third with cow MDE).

### 2.11. Isolation of Peripheral Blood Mononuclear Cells

Blood was collected into preservative-free heparin tubes and peripheral blood mononuclear cells (PBMCs) were isolated by centrifugation over Lymphoprep (Nycomed, Sheldon, Birmingham, UK). Diluted blood in an equal amount of PBS (2% FCS) was added to Lymphoprep and the mix was centrifuged at 650× *g* for 20 min with the break off. The upper plasma phase was discarded and the interface containing PBMCs was diluted in PBS and further centrifuged at 650× *g* for 10 min. Accordingly, PBMCs were diluted in RPMI medium supplemented with 10% fetal calf serum (FCS) (Biological Industries, Beit Haemek, Israel) and cultured in vitro. Following 24 h of incubation, cells were pelleted and Trizol reagent (INVITROGEN, Carlsbad, CA, USA) was added for RNA isolation.

### 2.12. Statistical Analysis

Statistical analyses of the data were performed using Prism 6.0 by employing the nonparametric Mann–Whitney test and a *t*-test. All results were expressed as mean ± the standard error of the mean (SEM). All graphs were constructed in Prism 6.0 (GraphPad Software Inc, La Jolla, CA, USA).

### 2.13. Ethical Approval Information

This study was approved by the Investigational Review Board (IRB) of Hadassah-Hebrew University Hospital (HM0-0101-13).

The protocols of this study in DSS-induced colitis model in mice were approved by the Ethics Committee—research number: MD-20-15923-4.

## 3. Results

### 3.1. Isolation and Characterization of Exosomes in Cow and Human Milk

Vesicles isolated from cow and human milk, identifiable as exosomes, were shown to be similar (Figure 1). Analysis by transmission electron microscopy showed that nanovesicles isolated from cow and human milk had a typical round- or cup-shaped appearance (Figure 1A,F). We also applied other methodologies to confirm the size of the vesicles isolated from milk. For instance, we employed nanoparticle tracking analysis (NTA), a conventional method used in characterizing exosomes, to measure the size of vesicles based on the tracking of the Brownian movement. The mean size was observed to be 171 nm for vesicles isolated from cow milk and 159 nm from those isolated from human milk (Figure 1B,G). We also applied another technique, the dynamic light scattering (DLS) method, to measure the size of vesicles using the Z-average that was shown to be 135.2 nm and 96.91 nm for vesicles isolated from cow and human milk, respectively. In addition, this technique allowed for the evaluation of another crucial parameter, the polydispersity index (PDI), used to characterize the size distribution of exosomes. The PDI was demonstrated to be 0.281 for vesicles isolated from cow milk and 0.261 for those isolated from human milk (Figure 1C,H), revealing a relatively even size distribution of exosomes, which could also be confirmed by the sharp single peak in our NTA analysis. According to these results, the vesicles isolated from cow and milk were demonstrated to have the characteristic size of exosomes. The purity of exosomes was assessed by western blot analysis. As shown in Figure 1D,I, all exosomes expressed the CD9 and CD81 exosome-related proteins, whereas heat-shock protein 70 (HSP70); non exosomal protein was only detected in total cellular lysates and not in exosomes, indicating that the isolated exosomes were highly purified and not contaminated by redundant intracellular components and debris (Appendix A). As mentioned, miRNAs are one of the main cargos of exosomes. Indeed, we isolated several miRNAs from both cow and human milk (Figure 1E,J). Taken together, these results demonstrated that exosomes were successfully isolated from cow and human milk with high purity and were further efficiently characterized using various methods.

### 3.2. Uptake of MDEs in vivo

Mice were monitored daily for 7 days during gavage administration of MDEs, and no signs of illness, intolerance, or weight loss were noted (Figure 2A). Milk-derived exosomes from cow and human milk were labeled with an infrared fluorescent membrane dye, namely DiR dye, to track their localization patterns in vivo. Accordingly, imaging revealed an accumulation of the fluorescent signal in the intestine of mice, following administration of cow and human MDEs (Figure 2B,C).

### 3.3. Oral Administration of MDEs Isolated from Cow and Human Milk Attenuated the Severity of Colitis Induced by DSS

Colitis was induced in Balb/c mice using 5% DSS provided for 1 week in their drinking water. Balb/c mice were treated for 6 days with (EXO+) or without (EXO−) exosomes, as control. During DSS treatment, the mice loss about 8 to 14% of the original weight (non-significant). Following the end of the treatment animals started to regain weight and they reached the initial weight at the end of the experiment. However, the MDE treated mice started earlier to gain weight (Figure 3A,B). As can be noted in Figure 3C,D there was a significant difference at day 2 (*p* < 0.01).

Treatment with MDEs was demonstrated to also significantly reduce the shortening of the colon (*p* < 0.05) (Figure 3E,F). Following treatment with cow MDEs, the median length of the colon was observed to be 7.5 cm compared with 6 cm, which was the median length of the colon in the control group (Figure 3E). Following treatment with human MDEs, the median length of the colon was shown to be 7.7 cm compared with 7 cm, which was the median length of the colon in the control group (Figure 3F). Moreover, cow and human MDE exhibited the beneficial effects of diminishing the signs and symptoms of DSS-induced colitis. Exosomes isolated from cow and human milk attenuated the severity of colitis induced by DSS, as shown by the pathological score. In particular, cow milk exosomes were shown to significantly reduce the pathological score from 5.83 ± 1.47 to 0.6 ± 0.6 (*p* < 0.05) (Figure 4B). Similarly, exosomes isolated from human milk significantly reduced the pathological score from 5.9 ± 0.58 to 4.13 ± 0.4 (*p* < 0.05) (Figure 4E).

Untreated colitic mice had extensive colonic damage and immune cell infiltration (in the lamina propria and mucosa), in comparison with MDE-treated colitic mice (Figure 4A,D). This was translated into a significantly higher histology score (*p* < 0.05) (Figure 4B,E) that comprised a higher score of inflammation and lymphocyte infiltration (Figure 4C,F). These results were in accordance with the colon shortening observed during colitis (see Figure 3E,F) and the lesion distribution shown in Figure 4G,H.

### 3.4. Effect of MDEs on Highly Expressed miRNAs and Their Gene Targets

Highly expressed miRNAs derived from milk [4] were noted to be the most abundant in the colitic colon of MDE-treated mice (Figure 5A). The four miRNAs analyzed in this study were part of 11 highly expressed milk miRNAs identified in our previous studies; miRNA-320, miRNA-375, let-7a, and miRNA148 [4]. The expression of miRNA-320, miRNA-375, and let7a was shown to be significantly higher in MDE-treated mice compared with untreated (Figure 5A). The expression of miRNA-148 was also noted to have a trend to be higher in MDE-treated mice compared with untreated (not significant). The target genes of the 11 highly milk miRNAs are shown in Figure 5B and Appendix A. The gene expression of DNMT1 and DNMT3, which are targets of the highly expressed milk miRNAs, were observed to be significantly downregulated in the colon tissue following treatment with MDEs (*p* < 0.05) (Figure 5C,D).

### 3.5. Milk-Derived Exosomes Affected Inflammation-Related Genes

We observed a lower mRNA expression of the interleukin 6 (IL-6) and tumor necrosis alpha (TNF-α) proinflammatory cytokine genes in the MDE-treated compared with the untreated group. In the colon of MDE-treated mice the mean level of IL-6 was shown to be 0.78 ± 0.32, whereas in untreated mice it was 3.82 ± 1.82 (Figure 6A). Likewise, in the colon of MDE-treated mice the mean level of TNF-α was demonstrated to be 0.86 ± 0.28, whereas in untreated mice it was 1.59 ± 0.24 (Figure 6B). Moreover, the expression level of IL-6 was significantly lower on peripheral blood mononuclear cells (PBMCs) isolated from blood of treated mice compared with that in untreated. The expression of IL-6 in PBMCs of MDE-treated mice was 0.573 ± 0.026, whereas it was 1 ± 0.7 in PBMCs of untreated mice (Figure 6C).

### 3.6. Milk-Derived Exosomes Increased the Level of TGF-β1

As mentioned, TGF-β is one of the cargo proteins of MDEs. We detected TGF-β1 as one of the cargo proteins of cow and human MDEs (Figure 7A). Accordingly, TGF-β1 was found to be highly expressed in MDEs compared with other milk fractions, such as the pellet following ultracentrifugation at 40 K (Appendix A). Following treatment with MDEs, we observed a significant increase in the protein level of TGF-β1 in the colon tissue, from 0.433 ± 0.07 in the colon of untreated to 1 ± 0.08 in MDE-treated mice (*p* < 0.05) (Figure 7B,C).

## 4. Discussion

MDEs might represent a naturally occurring new therapeutic modality for the management of IBD. There have been precedents regarding our observation that MDEs could slow down inflammatory processes. For example, Arntz et al. found a positive effect of MDEs in a murine arthritis model. Both cartilage pathology and inflammation were shown to be decreased following administration of MDEs [8]. Few studies, however, have attempted to explore the role of MDEs on IBD. Wu et al. demonstrated that the depletion of MDEs and their miRNA cargos exacerbated inflammation in a murine model of IBD [19]. In our study, we used DSS-induced colitis in mice treated with MDEs isolated from human and cow milk. Accordingly, we noticed that MDEs isolated from human and cow milk significantly reduced colon shortening (Figure 3E,F), and the pathological score of inflammation and mucosal damage (Figure 4). We observed that in MDEs treated mice there is a stop in the weight loss before untreated mice, (only significant at the second day of the treatment), meaning that the MDE accelerate the weight recover. In mice treated with human MDEs, the effect in the weight loss is less noted. Moreover, MDEs were shown to downregulate the inflammatory process in part through the downregulation of key cytokines involved in colitis, such as IL-6 and TNF-α (Figure 6). Accordingly, IL-6 was found to be elevated in PBMCs in patients with IBD [20]. Indeed, we found that IL-6 was downregulated in PBMCs isolated from mice treated with MDEs. Of note, IL-6 is considered to be a target gene of two highly expressed miRNAs contained in MDEs; let-7a and miRNA-148 [21,22]. We found that these miRNAs were upregulated in the colon tissue following treatment with MDEs, suggesting that they might play an immunosuppression role via regulation of IL-6. IL-6 is a major proinflammatory cytokine in various inflammatory diseases. Indeed, elevated Il-6 levels were detected in rheumatoid arthritis [23], ulcerative colitis and Crohn’s disease [24,25]. The concentration of IL-6 was shown to be significantly increased in IBD patients compared to healthy control and was shown to be in correlation with the disease activity [26]. IL-6 serum levels can be used as a prognostic marker in colitis. Moreover, IL-6 is a predictor for therapeutic effectiveness [27,28].

One of the main findings of our study was that that highly expressed milk miRNAs, such as miRNA-375, let-7a, miRNA-148 and miRNA-320 were significantly more abundant in the colon of MDE-treated mice (Figure 5A). It has been indicated that miRNAs might play a role in the therapeutic effect of MDEs in colitis by regulating the expression of target genes. For example, miRNA-320 was significantly decreased in the mucosa of pediatric patients with IBD [29]. The expression of miRNA-320 has been negatively correlated with the expression of nucleotide binding oligomerization domain containing 2 (NOD2), the first gene that was associated which increased susceptibility to Crohn’s disease. Interestingly, miRNA-375 and miRNA-320, which are known to be downregulated in colitis, were demonstrated to be highly expressed in the colon of colitic mice following treatment with MDEs (Figure 5A).

We found that DNMT1 and DNMT3, two of the most common target genes of miRNAs highly expressed in milk (Figure 5B), were downregulated in the colonic tissue following treatment with MDEs (Figure 5,C). It is known that DNMTs are key regulators of genes controlling methylation as part of the epigenetic process. Accordingly, aberrant DNA methylation patterns have been observed in many chronic inflammation conditions, such as IBD [30]. Several compounds targeting the DNA methylation status, such as folate, have been demonstrated to have a therapeutic effect on IBD. Therefore, our finding of the effect of MDEs on key methylation genes could be one of the mechanisms through which MDEs exert their therapeutic effect on colitis. Moreover, the levels of DNMTs have been reported to be increased in colorectal associated cancer (CAC) [31]. Noted, IBD might confer a predisposition to malignant transformation of normal epithelial cells [32]. Respectively, DNA modifications, such as alterations of their methylation patterns might directly affect the expression of genes that could have important implications in CAC. Based on the regulative effect of MDEs on the expression of DNMTs, we assumed that MDEs might have the potential to be used as a preventive anticancer therapy in patients with IBD.

Milk-derived exosomes are known to carry crucial signal components, such as regulatory proteins, for instance TGF-β. Several studies have also confirmed that bioactive TGF-β is carried by milk extracellular vesicles [16]. We indeed found consistent levels of TGF-β in MDEs isolated from cow and human milk. Dysregulation of the signaling of TGF-β has been observed in the intestine of patients with IBD. Clinical studies have shown the efficiency of TGF-β as a target in the treatment of IBD [33]. Moreover, enteral nutrition regimens containing TGF-β have been widely used as part of the enteral feeding for the treatment in patients with IBD. However, despite these formulas containing TGF-β they might not be able to delivery TGF-β to the intestine because of the degradation effect by intestinal proteolytic enzymes. In contrast, MDEs carrying TGF-β protect them from degradation and deliver them to the target in the intestine. Indeed, we found that following treatment with MDEs, the level of TGF-β protein was upregulated in the colon tissue and might play a role in the therapeutic effect of MDEs. These results were in accordance with previous studies that found that TGF-β in milk provided protection against inflammation in mice [34]. The implications of these results could establish MDEs as a plausible source of bioactive TGF-β that might be part of the entirety treatment to patients with IBD.

We previously showed that MDEs induced the proliferation of normal epithelial colon cells in a miRNA–dependent manner. Therefore, we hypothesized that MDEs might not only have an immune-regulatory function, but also a role in the proliferation of colonic epithelial cells [6]. By induction of the proliferation of epithelial cells, MDEs might also regulate the repair of colon tissue damage during colitis.

Based on the outcome of our previous studies on similarities in the expression of miRNAs in cow and human milk, we noted that MDEs isolated from both cow and human milk exert a similar effect on inflammation and tissue repair (Figure 3 and Figure 4). Moreover, in this study we also identified similarities in MDEs regarding their regulatory proteins’ cargo, such as TGF-β (Figure 7). Obviously, it would be more practical and commercial to isolate MDEs from cow than from human milk. Based on their functional similarities, cow milk could be a potential source of MDEs for use in human.

## 5. Conclusions

In summary, we have shown that human and cow MDEs are up taken by intestine cells, exerting a therapeutic and anti-inflammatory effect in a colitis murine model. We showed that treatment with MDEs involved several complementary pathways constituting the overall mechanism: regulation by miRNAs or proteins, such as TGF-β or DNMTs. Our findings open a new area for further studying the mechanisms underlying the effect of MDEs on immune-related diseases. The therapeutic effects of MDEs might thus have significant implications for the possible addition of MDEs as a nutrient in enteral nutrition formulas.

## Figures and Tables

**Figure 1 nutrients-12-02589-f001:**
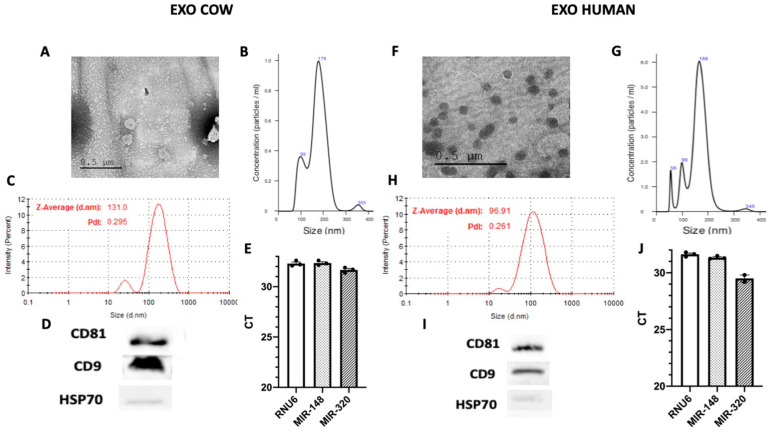
Isolation of exosomes from cow and human milk. Milk derived exosomes (MDEs) were isolated from cow (EXO COW) and human (EXO HUMAN) milk by sequential centrifugation. MDEs were analyzed using transmission electron microscopy with negative staining (**A**,**F**). The profile of the size distribution of MDEs was investigated using NanoSight. (**B**,**G**) The distribution of the particle size (by intensity) of isolated MDEs was determined by dynamic light scattering (DLS) (**C**,**H**). The protein expression of the CD81 and CD9 exosome-specific markers was evaluated via western blotting, with HSP70 used as a negative control (**D**,**I**). The expression of RNU6 miRNA-148 and miRNA-320 in MDEs was measured by qRT-PCR. The results of qRT-PCR analysis are shown as Ct values (**E**,**J**). Pdl = Polydispersity Index, CT = cycle threshold.

**Figure 2 nutrients-12-02589-f002:**
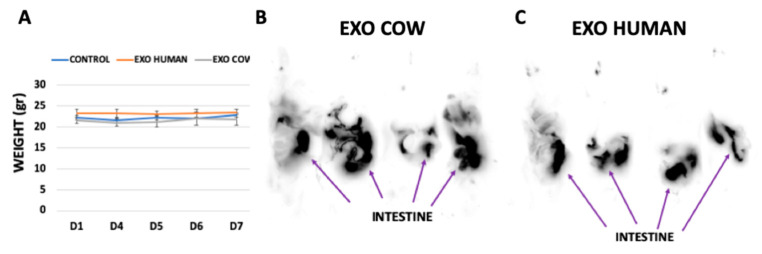
Exosomes are absorbed by the intestine. Balb/c mice were gavaged-administered exosomes isolated from cow and human milk for 7 d. (50 mg/kg in 200 μL Phosphate-buffered saline (PBS)). Mice were weighted during the whole period of the administration of exosomes (**A**) DiR dye-labeled MDEs isolated from cow (EXO COW) and human (EXO HUMAN) milk were administrated by gavage from the sixth day onward. Images were obtained for fluorescent analysis using the Typhoon FLA 9500 scanner (**B**,**C**), *n* = 4.

**Figure 3 nutrients-12-02589-f003:**
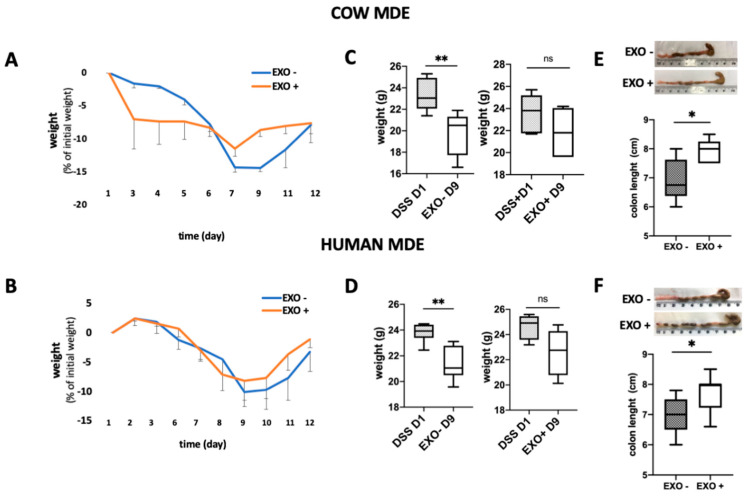
Effects of cow and human MDEs on a dextran sulfate sodium (DSS)-induced colitis model in mice. MDEs isolated from cow (EXO COW) and human (EXO HUMAN) milk were orally administrated to mice with DSS-induced colitis. The body weight of colitic mice (7 d of DSS (DSS)) with (EXO+) or without (EXO−) treatment with MDEs (**A**,**B**). Weight of DSS-treated mice at day 1 of the treatment with DSS (DSS D1) and at day 9; 7 d of treatment with DSS followed by 2 d with (EXO + D9) or without (EXO − D9) administration of MDEs (**C**,**D**). Colon length of colitis induced mice with (EXO+) or without (EXO−) treatment with MDEs. Images of representative colons from MDE-treated and untreated colitic mice (**E**,**F**), *n* = 5. Data are mean ± SEM. * *p* < 0.05, ** *p* < 0.01, ns = not significant (Mann–Whitney test).

**Figure 4 nutrients-12-02589-f004:**
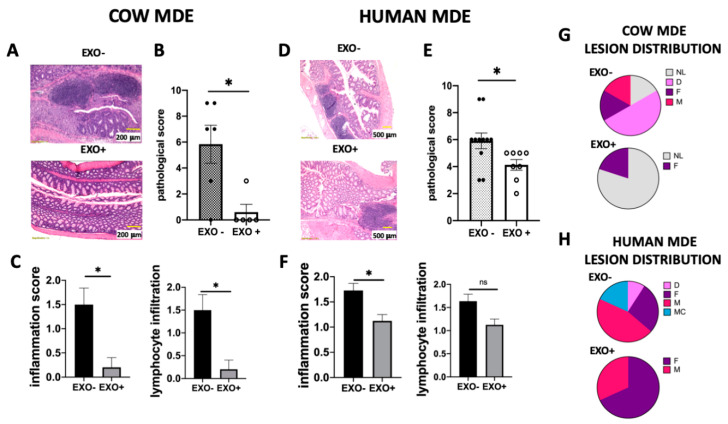
Oral administration of MDEs isolated from cow and human milk attenuated the severity of colitis induced by DSS. MDEs were orally administrated to Balb/c mice with DSS-induced colitis. MDEs isolated from cow and human milk were administrated by gavage for 5 d (COW EXO (A–C) and HUMAN EXO (D–F)). Representative microscopy images of hematoxylin and eosin (H&E)-stained colonic sections from MDE-treated and untreated mice (**A**,**D**). Histological scoring was performed on colon sections from mice treated (EXO+) or not (EXO−) with MDEs (**B**,**E**). Inflammation scoring, and lymphocyte infiltration were performed on colon sections from mice treated (EXO+) or not (EXO−) with MDEs (**C**,**F**). Distribution pattern of lesions in the colon of MDE-treated and untreated mice. NL: no lesion, D: diffuse, F: focal, M: multifocal, MC: multifocal to coalescing (**G**,**H**). *n* = 5. Data are mean ± SEM. * *p* < 0.05, ns = not significant, (Mann–Whitney test).

**Figure 5 nutrients-12-02589-f005:**
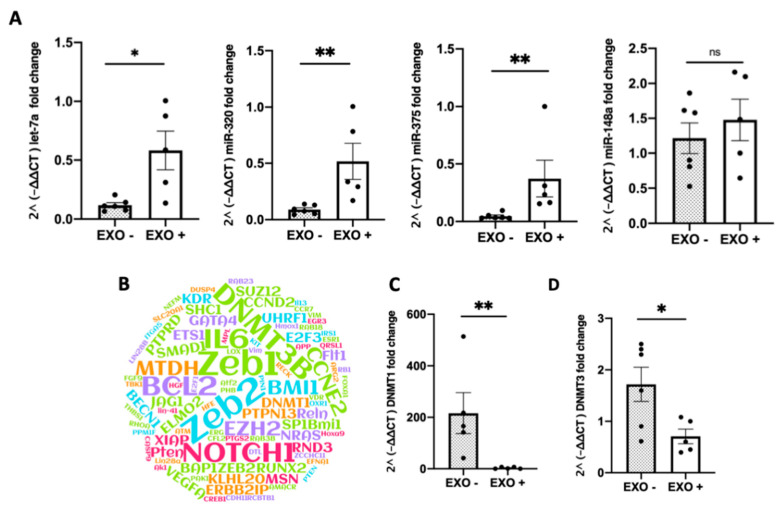
Oral administration of MDEs increased the content of highly expressed milk miRNAs in the colon of colitic mice following treatment with MDEs. Expression of miRNAs in the colon of colitic mice treated or not with MDEs. Expression of 4 of 12 highly expressed miRNAs derived from milk, namely let-7a, miR-320, miR-375, and miR148a by RT-PCR analysis. Obtained qRT-PCR values were calculated using the 2^ (-∆∆ CT) method, with values being normalized against the expression of RNU6B (**A**). Representation of the top 10 target genes known to be regulated by the 11 most abundant milk miRNAs. Target genes were weighted according to the rank of miRNAs in milk (MiRTarBase). The 11 most highly expressed miRNAs were determined by previous next generation sequencing (NGS) results in RNA isolated from whey fractions of human milk [4] (**B**). Expression of DNMT3 and DNMT1 in the colon of MDE-treated (EXO+) and untreated (EXO−) colitic mice. The expression level of DNMT3 and DNMT1 was analyzed by qRT-PCR. Values were calculated using the 2^ (-ΔΔ CT) method, and normalized against the expression of GAPDH (**C**,**D**). Data are mean ± SEM. *** p* <0.01, * *p* ≤ 0.05, ns = not significant. *n* = 5 (Mann–Whitney test).

**Figure 6 nutrients-12-02589-f006:**
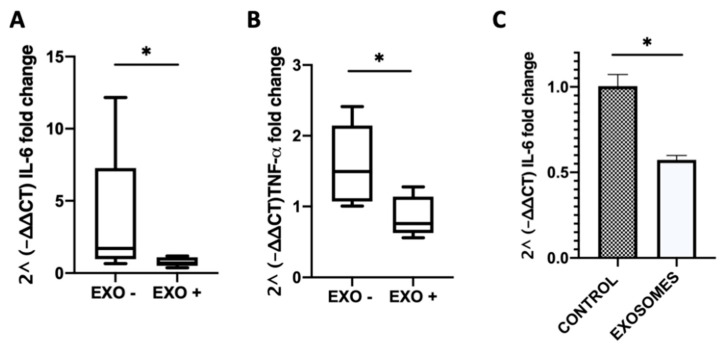
MDEs reduced the colitis-related expression of cytokines. Colitis was induced in Balb/c mice using 5% DSS provided for 1 wk in the drinking water. Balb/c mice were treated for 5 d with 0.5 mg/mouse in 200 μL PBS (EXO+) or PBS as control (EXO−). The expression level of IL-6 and TNF-α in the colon of mice treated with MDEs or PBS was analyzed by qRT-PCR Values were calculated using the 2^ (-ΔΔ CT) method, normalized against the expression of GAPDH (**A**,**B**). Data are mean ± SEM. * *p* < 0.05 (Mann–Whitney test). Expression of interleukin 6 (IL-6) in peripheral blood mononuclear cells (PBMCs) isolated from blood of colitic MDE-treated and untreated mice analyzed by qRT-PCR (**C**). Values were calculated using the 2^ (-ΔΔ CT) method, normalized against the expression of TBP. Data are mean ± SEM. * *p* < 0.05 (*t*-test).

**Figure 7 nutrients-12-02589-f007:**
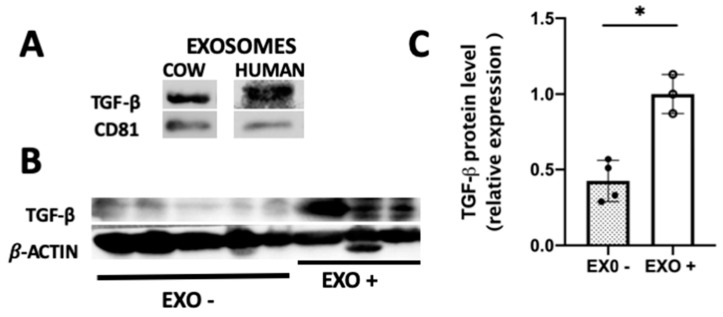
Oral administration of MDEs increased the content of the TGF-β1 protein. Expression of TGF-β1 protein in cow and human MDEs (**A**). Expression of TGF-β1 in the colon of mice following treatment with MDEs or PBS. Protein expression determined by western blotting, with β-actin used as loading control (**B**). Quantification was performed using the NIH-Image software (http://rsb.info.nih.gov/nih-image/download.html) (**C**); * *p* < 0.05.

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
