# Peer review of "Cow and Human Milk-Derived Exosomes Ameliorate Colitis in DSS Murine Model"

_nutrients, 2020, doi:10.3390/nu12092589_

Round 1

Reviewer 1 Report

This is an interesting study aimed at investigate the putative role of milk-derived exosomes as a potential therapeutic target in a murine model of inflammatory bowel disease.

The paper is well-written and clear; moreover, the similarity of results between human and cow-derived milk gave the opportunity of a more practical continuation of these studies.

In my opinion, the authors should discuss extensively the role of IL-6 in IBD murine model as well as in human in the discussion section, since the anti-inflammatory effect of MDEs seems to be related with the action of this  cytokine. Of note, IL-6 serum levels seem to be important even in predicting therapeutic effectiveness to biological therapies (i.e. Sato, J Gastroenterol Hepatol, 2015 ; Nishida, Scand J Gastroenterol, 2018 ; Bertani, Br J Clin Pharmacol, 2020).

Author Response

According to the suggestion of the reviewer we discusses more extensively the role of IL-6 in IBD, their importance as therapeutic maker an in predicting effectivenesses to therapies. See discussion section, line 407-411.

Reviewer 2 Report

The current manuscript by Reif, S., et al. studies the potential of using cow and human milk-derived exosomes (MDE) for the treatment of inflammatory bowel disease (IBD). The authors studied the effects of MDE in a mouse model of dextran sulfate sodium (DSS)-induced colitis by measuring weight loss, histopathological scoring, colon length, inflammatory markers, and various miRNA and protein levels. In brief, treatment with MDE appears to correspond with the recovery of mice from colitis, thus implying the use of MDEs as potential therapeutics. While the manuscript might present good data and results, its current form makes it difficult to follow and analyze the results in a way appropriate for stringent peer-review. Hence, the entire manuscript needs major revisions in writing, formatting, and structuring the results.

The title of the manuscript “Cow and human milk-derived exosomes as a therapeutic target for inflammatory bowel disease” needs to be reanalyzed, especially the word, therapeutic target.

I recommend the introduction be significantly improved to include information on IBD (e.g., what diseases are included under the IBD, possible pathogenesis, risk factors, and any statistical data available on the effect of age/sex/race on the risk of developing IBD), current treatment options and their limitations, followed by information on exosomes, MDE, the benefits and/or rationale for testing them.

In the Materials and Methods section, the authors provide reasons for why a particular method was chosen. Such reasoning and/or information should be moved to the Results or Discussion section, and the Materials and Methods section be limited to just explaining how the method is performed in detail. Information on how many times each experiment was performed is also required.

Further, the authors need to restructure the results section to follow the sequence of the figures. For example, the authors discuss Figure 4 before discussing Figure 3 (lines 235-247), Figure 6B before discussing Figure 6A (lines 262 and 264). Importantly, the figures are too small to read and analyze and need to be improved both in size and pixels. I suggest (and do not mandate) the authors provide full gel images as supplementary information or only to reviewers, to better understand the data. Gels should also show controls – the use of actin as control was mentioned in the Methods section. This needs to be shown in figures as well. I also recommend placing the figures where appropriate instead of grouping them all in two pages, e.g., just below or above the text, for the convenience of the readers.

The above points cover most necessary changes and a few specific comments are listed below.

Line 9: In the abstract, exosomes isolated from cow and human milk are abbreviated as MDEs. Please replace it with the correct full form of the abbreviation

Line 10: What are MDE’s labeled with?

Line 14: “The weight loss was reverted in MDE-treated mice” needs better explanation

Lines 15, 16, 17, 18: It is important to give the reader a general understanding of the roles of each of these proteins and miRNAs. Are they inflammatory? Are they beneficial? If so, how?

Line 37: “Based on our preliminary results, we suggested that milk-derived exosomes (MDEs) could be used 37 to treat colitis.” Are these results published somewhere? If yes, please cite it. If not, please elaborate on the preliminary experiments and results.

Line 50-52: Could the authors elaborate their hypothesis for how treatment with MDE’s might restore normal intestinal microbial flora?

Line 148: How many mice were included in each group? Why is treatment with DSS stopped, and not continued along with MDE or control treatment? Wouldn’t this type of model be very different from a natural setting where the progression of the disease continues and not stop after a few days?

Line 156: Is there a reason why mice were dosed the same amount of 0.5 mg/mouse irrespective of their initial body weights? Typically, preclinical studies dose in mg/kg considering the weight differences in animals.

Line 173: PBMC is used without explanation or a full form.

Line 227-229 and Figures 3A and 3B: Weight loss data represented in its current form makes it difficult to tell any difference between control and MDE treated mice. Based on the current figure, it appears to me that stopping DSS is sufficient for mice to regain weight and doesn’t depend on MDEs at all. How did the authors calculate these differences? Why is AUC taken into consideration? Could the authors re-graph the data into a single line graph (e.g., like Figure 2A) showing differences between control and MDE treated mice? Also, in figure 2B, there doesn’t seem to be any decrease in the weight in DSS treated mice.

Figures 3C and 3D: Why do the authors consider day 2 after the start of MDE treatment to show the differences rather than at the end of the study?

Lines 234-237: The data is missing “±”.

Line 244-247: Please point out the exact panel in each figure rather than the whole figure.

Line 253-254: Since the statistics show non-significance, there is no difference between miRNA-148 levels. Please change the wording accordingly.

Figure 5B: I recommend writing down the target genes in text or including it as a table with target genes for each miRNA. The figure in its current form is difficult to comprehend.

Figure 6B: Is there a significant difference between the two groups?

Line 269: Why is Figure 7A corresponding to the text “TGF-β is one of the cargo proteins of MDEs”?

Lines 342-363: First three paragraphs of the Discussion section are more appropriate in the Introduction.

Author Response

We thank the reviewer for reviewing our paper and for the beneficial comments. 

We modified the manuscript title according to the reviewer's suggestion to: “Cow and human milk-derived exosomes ameliorate colitis in DSS murine model”.
2. According to the reviewer's suggestion, we now included in the revised manuscript information about IBD (Introduction line 27 to 35).
3. According to the reviewer's suggestion, we excluded from the M&M section the reason why a particular method was chosen. We now included the number of experiments in the methods section (lines 182-3 and 204-5).
4. We restructure the results section according to the reviewer's suggestion. We improved the size and quality of the figures. (The figures are also uploaded separately in a high quality format)The full gel image is now provided. (Figure 7B). The figures are placed in two pages based on the template of the journal requested.
5. The full form of MDE was replaced with the correct full form. (Abstract: line 9-10)
6. Abstract line 10: now it is added that MDEs were florescent labeled.
7. As has been requested by the reviewer we modified line 14 in the abstract.
8. “Lines 15, 16, 17, 18: It is important to give the reader a general understanding of the
roles of each of these proteins and miRNAs. Are they inflammatory? Are they
beneficial? If so, how?”

Due to the limited space (Abstract: 200 words), We could not add more information in the Abstract.
9. “Line 37: “Based on our preliminary results, we suggested that milk-derived exosomes (MDEs) could be used 37 to treat colitis.” Are these results published somewhere? If yes, please cite it. If not, please elaborate on the preliminary experiments and results.”
Indeed, the reviewer is very right. This sentence is irrelevant and was put by our
mistake. We have no previous publication in this subject.
10. We included information about the effect of milk extracellular vesicles on the intestinal microbial flora in the Introduction section (line 84-85)
11. “Line 148: How many mice were included in each group? Why is treatment with DSS stopped, and not continued along with MDE or control treatment? Wouldn’t this type of model be very different from a natural setting where the progression of the disease continues and not stop after a few days?”
Similar to IBD in DSS induced colitis, it is not necessarily that the trigger will continue during all the disease. The DSS model is a well-established model for colitis studies.
This model is based on the DSS given in drinking water for 5-7 days to induce the epithelial damage that initiates the inflammatory response that induces colitis.
Following the induction of the colitis, the DSS trigger is stoped and the therapeutic treatment was initiated.The number of animals/group were added (legend of figure 3 and 4).
12. The initial weight of the mice was similar and the changes in the weight could not affect significantly the volume of the MDE dose administration. Indeed, the reviewer is very correct and the dose must be expressed by mg/kg. By mistake the dose was written as 5 mg/mouse instead of 0.5 mg/mouse (20g) (corresponding to 25mg/kg). According to the review comment we put the dose in the correct form (line 189).
13. We now included the full form of PBMC. (M&M line 207-8).
14. “Line 227-229 and Figures 3A and 3B: Weight loss data represented in its current form makes it difficult to tell any difference between control and MDE treated mice. Based on the current figure, it appears to me that stopping DSS is sufficient for mice to regain weight and doesn’t depend on MDEs at all. How did the authors calculate these differences? Why is AUC taken into consideration? Could the authors re-graph the data into a single line graph (e.g., like Figure 2A) showing differences between control and MDE treated mice? Also, in figure 2B, there doesn’t seem to be any decrease in the weight in DSS treated mice.”

We changed the graph according to the reviewer's suggestion (Fig 3A and 3B). It can be noted that there is a trend of weight restore in cow MDEs treated mice compared to the untreated group. In mice treated with human MDEs the effect in the weight loss is less noted. The weight loss in MDE treated mice was significant only on day 2 following treatment compared to untreated mice. This point is now clarified in the results section (lines 268-271) and in the discussion section (lines 397-400).
15. Figures 3C and 3D: Why do the authors consider day 2 after the start of MDE treatment to show the differences rather than at the end of the study?
In contrast to the inflammatory parameters, the weight loss at the end of the experiment did not show significant changes as the reviewer indicated. Although, the inflammation and mucosal damage are still active at this time point (Figure 4). The weight loss is one of the parameters that can indicate the activity of the disease but as was a comment by the reviewer the weight loss is reverted following the stooping of DSS. In our work, we observed that in treated mice there is a stop in the weight loss before untreated mice, (only significant at the second day of the treatment), meaning that the MDE accelerates the weight recover. This point is now clarified in the discussion section (line 397-400).
16. We corrected the spelling typo and including now the missing of ± in all the manuscript.
17. The exact panel in each figure is now pointed in the text of the results section 1.3, 3.1.4., and 3.1.5.
18. There is a trend in the effect of MDE on miR-148 expression. We emphasize now that there is a trend and not a change. (results line 295)
19. We improved figure 5B (target genes) to be more clear. Moreover, we added the list of the gene target of each of the 11 highly miRNA expressed in as an excel supplementary file (S3).
20. By mistake, we omitted that the difference in IL-6 gene expression is statistically significant. We added this information in figure 6 (Fig 6A).
21. The text in the results section, figure and figure legend related to the TGF were revised and corrected (legend of Figure 7 and results section (lines 311-2 and 316)).
22. Indeed the review is correct. We moved most of the content of the first three paragraphs from the Discussion section to the Introduction section.

Round 2

Reviewer 2 Report

The authors have addressed most of the points